# A Multifunctional Coating on Sulfur-Containing Carbon-Based Anode for High-Performance Sodium-Ion Batteries

**DOI:** 10.3390/molecules28083335

**Published:** 2023-04-10

**Authors:** Lin Zhu, Bo Yin, Yuting Zhang, Qian Wu, Hongqiang Xu, Haojie Duan, Meiqin Shi, Haiyong He

**Affiliations:** 1School of Chemical Engineering, Zhejiang University of Technology, Hangzhou 310014, China; zhulin@nimte.ac.cn; 2Ningbo Institute of Materials Technology and Engineering, Chinese Academy of Sciences, Ningbo 315201, China; zhangyuting@nimte.ac.cn (Y.Z.); wuqian20@nimte.ac.cn (Q.W.); xuhongqiang@nimte.ac.cn (H.X.); duanhaojie@nimte.ac.cn (H.D.)

**Keywords:** sodium-ion batteries, sulfur-containing anode, multifunctional carbon coating, shuttling effect, C-S covalent bond

## Abstract

A sulfur doping strategy has been frequently used to improve the sodium storage specific capacity and rate capacity of hard carbon. However, some hard carbon materials have difficulty in preventing the shuttling effect of electrochemical products of sulfur molecules stored in the porous structure of hard carbon, resulting in the poor cycling stability of electrode materials. Here, a multifunctional coating is introduced to comprehensively improve the sodium storage performance of a sulfur-containing carbon-based anode. The physical barrier effect and chemical anchoring effect contributed by the abundant C-S/C-N polarized covalent bond of the N, S-codoped coating (NSC) combine to protect SGCS@NSC from the shuttling effect of soluble polysulfide intermediates. Additionally, the NSC layer can encapsulate the highly dispersed carbon spheres inside a cross-linked three-dimensional conductive network, improving the electrochemical kinetic of the SGCS@NSC electrode. Benefiting from the multifunctional coating, SGCS@NSC exhibits a high capacity of 609 mAh g^−1^ at 0.1 A g^−1^ and 249 mAh g^−1^ at 6.4 A g^−1^. Furthermore, the capacity retention of SGCS@NSC is 17.6% higher than that of the uncoated one after 200 cycles at 0.5 A g^−1^.

## 1. Introduction

Over the past decade, the energy crisis, environmental degradation, and political orientation across the world have combined to fire the explosive growth of the new energy vehicle market [1,2]. However, limited by barren lithium resources, Lithium-ion batteries (LIBs) cannot support the rapid development of electric vehicles and large-scale energy storage alone [3]. The high crustal abundance, low standard reduction potential (−2.71 V vs. SHE), and inert nature of sodium when it encounters aluminum make sodium-ion batteries (SIBs) a cheaper and more widely available alternative to LIBs [3,4,5]. More importantly, the operating principles and battery components of SIBs are similar to LIBs, which means the advanced manufacturing line and technique of LIBs can be applied to the production of SIBs without resistance [2,3,5].

Currently, the biggest obstacle to the commercial production of SIBs is the lack of suitable electrode materials. Although various anode materials have been intensively studied, such as conversion materials (oxide [6], sulfide [7]), alloy materials (Sn [8], Sb [9], and Ge [10], etc.), organic compounds (Schiff base polymer [11] and polyamide [12]), and intercalation materials (carbon-based material [13] and titanium-based material [14], etc.), a carbon-based material is still the most promising one for its low cost, chemical inertness, adjustable structure, and abundant sources (Appendix A). For example, Kang et al. reported that graphite as a host for Na^+^-solvent complexes could provide a sodium storage specific capacity in excess of 120 mAh g^−1^ [15]. In 2000, glucose-derived hard carbon (HC) as the anode of SIBs was first reported by Dahn et al. and demonstrated a high reversible capacity of 300 mAh g^−1^ [16]. From then on, HC, as carbonaceous materials that cannot be converted into graphite even at temperatures above 3000 °C, has attracted intensive investigation [17]. HC materials with different interlayer distance, doping elements, and pore structure have been reported in succession [18]. However, the stress-induced process accompanying the insertion/desertion of sodium ions deteriorates the cycling stability of HC [19]. Additionally, the sluggish reaction kinetics during the insertion and pore-filling process make the rate capability of HC very poor [20].

An effective way to overcome the shortages of HC is to dope with heteroatoms such as B, N, P, and S [21]. Doped heteroatoms can optimize interlayer distance, electron/ion conductivity, porosity, and defect concentration of hard carbon material to increase the active sites for sodium storage [22,23,24,25]. For example, benefiting from the expanded interlayer distance (0.406 nm) and enhanced electronic conductivity of hierarchical N/S–coped carbon microspheres (NSC-SP), a NSC-SP electrode delivered a high specific capacity of 280 mAh g^−1^ at 30 mA g^−1^ and 130 mAh g^−1^ at 10 A g^−1^ [26]. Additionally, the introduction of heteroatoms can also enhance the rate capability of electrode material because of the increase in sodium storage capacity dominated by pseudocapacitance [21]. According to the report of Jin et al., the capacitive contributions of N, S co-doped carbon nanoparticles (NSC2) at 5 mV s^−1^ is 94%, which is higher than that of undoped carbon nanosheets (DCs, 86%). As a result, NSC2 exhibited higher discharge specific capacities at various current densities [27]. Among the common non-metallic doping elements, sulfur is preferred because sulfur atoms can exist not only in the covalent bonding state (C-S-C, C-S-S-C), but also in the molecular state of S_2_, S_4_, S_6_, and S_8_ stored in the pores of HC [28,29]. Each single sulfur atom can anchor two sodium ions during the discharge to significantly improve the sodium storage capacity of HC. However, some HC materials, such as glucose-derived carbon spheres (GCS), have difficulty in preventing the shuttling effects of soluble polysulfide intermediates when sulfur molecules are in the pores of GCS, deteriorating the cyclic stability of electrode materials. Fortunately, the research findings of Lithium-Sulfur batteries inspire us to solve this problem from a multifunctional carbon coating layer [27,30]. For example, Song [31] designed a polypyrrole coated loose mesh carbon/sulfur composite (C/S@PPy). In this structure, the conductive network of mesoporous carbon has good electrical conductivity, and the carbon coating enhances the electronic conductivity of the sulfur cathode. In addition, the coating inhibits the dissolution of polysulfide. The initial discharge capacity of the C/S@PPy electrode material was 1209.6 mAh g^−1^ at 0.1 C. The discharge capacity retention rate was 62.2% at 0.2 C for 300 cycles.

In this paper, a polyacrylonitrile derived N, S-codoped carbon layer (NSC) is coated on the glucose-derived and sulfur-loaded carbon spheres (SGCS@NSC) through a solvothermal reaction and subsequent thermal treatment. As a physical barrier layer, the NSC layer can confine polysulfide in the GCS to prevent its dissolution in the electrolyte. More importantly, the abundant polarized C-S and C-N covalent bonds in NSC have a strong anchoring ability to further eliminate the shuttling effect of polysulfide. As a result, SGCS@NSC exhibits a significantly improved cyclic stability, with a 17.6% higher capacity retention than SGCS after 200 cycles at 0.5 A g^−1^. Besides, SGCS@NSC inherits the cross-linked three-dimensional network structure of NSC, which can serve as a continuous conductive network to achieve excellent electrochemical performance. SGCS@NSC exhibits a considerable sodium storage capacity of 609.8 mAh g^−1^ at 0.1 A g^−1^ and excellent rate capability of 249 mAh g^−1^ at 6.4 A g^−1^. The synergistic effect of physical confinement and chemical adsorption can be used to improve the electrochemical performance of other sulfur-rich electrode materials.

## 2. Results and Discussion

Glucose-derived carbon spheres (GCS) of about 150 nm were firstly prepared according to the reports elsewhere (Figure 1a). After p-GCS were fully dispersed in the polyacrylonitrile (PAN) solution, the homogeneous suspension was titrated into the high-speed rotating mixed alcohol solution to encapsulate p-GCS into a cross-linked 3D PAN network. SGCS@NSC, NSC, and SGCS samples were finally obtained by annealing a mixture of the corresponding precursor and sublimated sulfur. As shown in Figure 1b,c and Appendix A, both SGCS@NSC and NSC maintain the cross-linked network structure that facilitates fast electron transport. The difference is that there are more sphere-like particles in SGCS@NSC due to the encapsulation of SGCS into the network. The TEM image in Figure 1d further reveals the detailed structure of SGCS@NSC and validates the conclusion drawn from SEM images. The HRTEM (Figure 1e) shows that SGCS particles are tightly encapsulated with NSC of uneven thickness. The NSC layer acts as a physical barrier to prevent the diffusion of soluble polysulfides from the SGCS particles to a certain extent. Energy dispersive X-ray spectroscopy (EDS) demonstrates the coexistence and uniform distribution of C, N, and S elements in SGCS@NSC. The sulfur element in the NSC layer in the form of C-S covalent bonding also plays an important role in eliminating the shuttling effects of polysulfides, which will be discussed later.

X-ray diffraction (XRD) patterns, nitrogen adsorption/desorption curves, Raman spectra, and XPS spectra were obtained to understand the structural characteristics and chemical properties of obtained samples. The XRD patterns of prepared samples all show broad (002) peaks around 25° corresponding to characteristics of amorphous carbon (Appendix A) [32,33]. The interlayer distance (d_002_) of SGCS@NSC, NSC, SGCS, and GCS samples are 0.361 nm, 0.352 nm, 0.367 nm, and 0.389 nm, respectively. The interlayer distance of SGCS is lower than that of GCS due to the low thermal treatment temperature and the promoting effect of molten sulfur on the uniformity of the thermal field. This also implicitly suggests that the sulfur species in SGCS are not primarily presented in the form of chemical bonds, a view that will be discussed in detail in conjunction with subsequent XPS characterization. On the other hand, it is completely reasonable that the interlayer distance of SGCS@NSC composite is between SGCS and NSC itself. Nitrogen adsorption/desorption tests indicate that GCS owns the highest Brunauer–Emmett–Teller (BET) surface area (56.1 m^2^ g^−1^, Appendix A). After the introduction of sulfur, the BET surface area of SGCS decreases to 18.7 m^2^ g^−1^, corresponding to a severe loss of pores between 10 and 50 nm (Figure 2a, obtained from the adsorption curve). Considering the BET surface of NSC is 31.2 m^2^ g^−1^, its reasonable that the BET surface area and pore volume exhibit the sequence of GCS > SGCS@NSC > SGCS.

The Raman spectra of SGCS@NSC and NSC show peaks at 175, 312, 375, and 804 cm^−1^ in the 100–1000 cm^−1^ wavenumber region, indicating the stretching, bending, and deformation of the C-S bond (Figure 2b and Appendix A) [33,34]. The stretching of the S-S bond contributes to the two peaks at 476 and 941 cm^−1^ [35]. In addition, the two peaks near 1480 cm^−1^ also correspond to the stretching of C-S/S-S bonds [33,36]. The characteristic peaks of C-S/S-S bonds in SGCS are hardly observed. Sulfur in SGCS is mainly stored in the pores as an independent molecular state, making C-S/S-S bonds difficult to detect on surface, and the few sites in GCS where sulfur can form covalent bonds cause the absence of C-S/S-S bonds. In the high wavenumber region, all the prepared samples show two main peaks: one centered at 1326 cm^−1^ corresponds to the D band caused by defects or disordered structures, and the other around 1580 cm^−1^ represents the G band generated by the in-plane stretching vibration of sp^2^ hybridized carbon [37,38]. The exact wavenumber of the D band of SGCS and SGCS@NSC are close to each other, while the G band of SGCS (1542 cm^−1^) displays a red shift compared to SGCS@NSC (1556 cm^−1^), which may be due to more edge defects induced by a larger interlayer distance of SGCS. The Raman characteristic peaks of SGCS@NSC are consistent with NSC, suggesting that SGCS particles are well wrapped by the NSC layer. The D-band to G-band intensity ratio of SGCS@NSC (1.203) is much higher than SGCS (0.937) and GCS (0.168), indicating N/S co-doping can introduce more structural defects, which is conducive to improving the sodium storage performance [39].

The high content of the C-S bond in SGCS@NSC is also reflected in XPS spectra. Appendix A confirm the coexistence of C, O, and S elements in SGCS, NSC, and SGCS@NSC, and the latter two also contain significant amounts of the N element. Figure 2c shows the fitted S 2p high-resolution XPS spectra of SGCS@NSC. The two main peaks at 162.7 and 164.0 eV are contributed by C-S and S-S bonds, respectively [27,36], and another peak at 160.8 eV can be assigned to HS_X_C- by-products produced by PAN during the vulcanization process [40]. According to the quantitative analysis, the ratio of the C-S bond to S-S bond in SGCS@GCS is 6.2, much higher than 1.9 in SGCS (Appendix A). As for C 1s spectra, the area percentage of C-N/C-S bonds in SGCS@GCS (18.0%) far exceeds the percentage of C-S in SGCS (12.5%, Appendix A). These two comparisons demonstrate that SGCS@NSC possesses much more C-S bonds, which can serve as anchoring sites of polysulfide. In addition, the abundant pyridine nitrogen (389.9 eV), pyrrole nitrogen (400.7 eV), and oxidized nitrogen (402.7 eV) in SGCS@NSC exhibit strong dipole–dipole electrostatic interactions with polysulfides, preventing polysulfides from shuttling (Appendix A) [41,42]. Therefore, the chemical adsorption of the NSC layer is helpful to enhance the cyclic stability of SGCS@NSC.

The effectiveness of the NSC coating layer is verified by the electrochemical performance in sodium ion half-cells. Figure 3a shows the initial five cycles cyclic voltammetry (CV) curves of SGCS@NSC in the range of 0.01–3.0 (V vs. Na/Na^+^) with a sweep speed of 0.1 mV s^−1^ [43]. After the first cycle, two pairs of peaks at 2.239/1.855 and 1.903/1.137 V in the subsequent curves correspond to the step-by-step redox reaction of sulfur [44,45], and the curves overlap well, indicating the excellent cycling stability of the electrode. However, the cathodic peaks current of SGCS decreases rapidly from the second cycle, and the redox peaks’ intensities are weaker than SGCS@NSC despite the higher sulfur content of SGCS (Appendix A).

As shown in Figure 3b, the initial discharge/charge specific capacity of SGCS@NSC, SGCS, and GCS at 0.1 A g^−1^ is 942.9/798.2, 928.1/661.3, and 417.1/226 mAh g^−1^, respectively. SGCS@NSC has the highest initial coulombic efficiency (ICE, 84.7%) due to the moderate BET surface area and the suppressed polysulfides dissolution. Although GCS has the largest interlayer distance, the high BET surface area and the lack of highly reversible active sites introduced by N, S co-doping resulted in an ICE as low as 54.2% for GCS. The 5th, 10th, and 30th galvanostatic charge and discharge curves of SGCS@NSC overlap each other (Figure 3c), exhibiting the same electrochemical phenomena as the CV curves. After 60 cycles at 0.1 A g^−1^, the discharge specific capacity of SGCS@NSC is 609.8 mAh g^−1^, far exceeding that of SGCS (164.9 mAh g^−1^) and GCS (215.4 mAh g^−1^). The significant improvement in sodium storage performance can be attributed to N, S co-doped active sites, which can not only act as sodium storage sites themselves, but also enhance the reversibility of polysulfide intermediates in SGCS. After 200 cycles at 0.5 A g^−1^ (Figure 3d), the discharge specific capacity of SGCS@NSC, SGCS, and GCS are 294.1, 105.1, and 148.3 mAh g^−1^, corresponding to the capacity retentions of 46.1%, 28.5%, and 74.1%, respectively (all the electrodes were tested at 100 mA g^−1^ for 5 cycles to fully activate the electrode materials, and the capacity retention was calculated based on the 6th discharge specific capacity). GCS has a better cycling stability but lacks sufficient active sites for sodium storage, so the specific capacity is lower than SGCS@NSC by up to 146 mAh g^−1^. The inability of SGCS to maintain the reversibility of the sulfur-containing active sites resulted in the capacity retention and specific capacity to be 17.6% and 189 mAh g^−1^ lower than SGCS@NSC, respectively. Benefiting from the synergistic effect of the physical barrier and chemical adsorption of NSC layer, SGCS@NSC displays high sodium storage specific capacity and enhanced cycling stability.

Meanwhile, Figure 3e presents the rate performance of SGCS@NSC, NSC, and SGCS measured at various current densities. At the current densities of 0.1, 0.2, 0.4, 0.8, 1.6, and 3.2 A g^−1^, the discharge specific capacity of SGCS@NSC are 818.6, 710.2, 674.8, 623.6, 536.4, and 407.9 mAh g^−1^. At 3.2 A g^−1^, the discharge profiles of SGCS@NSC show an obvious step-by-step sodiation phenomenon, implying the excellent electrochemical kinetics (Figure 3f). Even when the current density is increased to 6.4 A g ^−1^, a high specific capacity of 248.9 mAh g^−1^ can still be provided, much higher than the 39.1 mAh g^−1^ of SGCS and 72.9 mAh g^−1^ of GCS. In addition, when the current density is reset to 0.1 A g^−1^, the SGCS@NSCC electrode still provides a reversible specific capacity of 609.8 mAh g^−1^, exhibiting excellent high-rate cycling performance. These results indicate that the successful incorporation of S and N elements into the carbon framework can effectively improve the reactivity and electronic conductivity of GCS by producing external defects, and the resulting defects can enhance the transport rates of sodium ions. More contact sites are provided between the active material and the electrolyte, which is beneficial to the high-rate performance.

To explore the reaction kinetics of the prepared electrodes, CV curves at different sweep speeds of 0.1–1.0 mV s^−1^ were obtained (Figure 4a and Appendix A). The area of the closed CV curves represents the total charge storage for Faraday and non-Faraday processes [46], which are usually divided into capacitive-control and diffusive-control charge storage [33]. The contribution of the two charge storage modes can be calculated according to the following equations [47]:(1)$$i=av^{b}$$
(2)logi=blogv+loga
where *a* and *b* are variable parameters, v means the sweep speed, and i is the response current. When the b value is close to 0.5 or 1, the diffusion-controlled process or the surface capacitance-controlled process dominates the electrochemical reaction [48]. The value of b is determined by the slope of the log (i) vs. log (v) plot of the redox peaks. As can be seen from Figure 4b, the b value of peak A of SGCS@NSC is between NSC and SGCS, which are both below 0.7, indicating a diffusion-controlled process. However, the b value of peak C of SGCS@NSC is around 0.8, higher than that of NSC (0.61) and SGCS (0.57), manifesting a surface capacitance-controlled discharge process. To quantify the contribution of the diffusion and surface capacitance-controlled processes at various sweep speeds, Equation (1) can be rewritten as:(3)$$i=k_{1}v+k_{2}v^{0.5}$$
where *k*_1_ and *k*_2_ are constants [49]. Figure 4c shows that the contribution of pseudocapacitance for SGCS@NSC are 44, 49, 57, 65, 77, and 90% from 0.1 to 1 mV s^−1^, exceeding SGCS and NSC at various sweep speeds. This result is consistence with the optimal rate capability of SGCS@NSC, demonstrating the resistance to high current pulses.

The galvanostatic intermittent titration technique (GITT) was performed to investigate the Na^+^ diffusion coefficient in prepared electrodes during the discharge and charge process. The calculation based on Equation (S1) shows that SGCS@NSC inherits the electrochemical kinetics process of NSC with similar Na^+^ diffusion coefficient and variation trends. The average Na^+^ diffusion coefficient of SGCS@NSC and NSC is higher than SGCS in the whole process, and the difference in the discharge process reaches an order of magnitude (Figure 4d,e). As shown in Figure 4f, the Nyquist plots of all samples were composed of two parts: one is a depressed half circle, which is composed of two half-circles in the high and middle frequency areas; and the other is an oblique linear in the low frequency area. Furthermore, the corresponding equivalent circuit model in Figure 4f is used to simulate the experimental data, where R_S_ stands for the resistance of electrolyte solution obtained from the high frequency region data and R_ct_ represents the charge transfer resistance fitted from the intermediate frequency region data [50]. The electrochemical impedance spectroscopy (EIS) results illustrate decreased semicircle diameters at high-frequency regions and increased slopes of the inclined lines at low-frequency regions for SGCS@NSC. As we can see from the fitted EIS parameters in Appendix A, the R_s_ (4.9 Ω) and R_ct_ (637.2 Ω) values of SGCS@NSC are significantly reduced compared to other samples. Profiting from the cross-linked conductive network constructed by NSC coating, the charge transfer between the SGCS@NSC electrode and the electrolyte is easier. Thus, it is demonstrated that the electron and sodium ion transfer rates in SGCS@NSC are both improved [50]. The fast Na^+^ diffusion coefficient and electron transport in the 3D cross-linked NSC coating layer account for the high sodium storage specific capacity and excellent rate capability of the SGCS@NSC electrode.

## 3. Materials and Methods

Synthesis of glucose-derived carbon spheres (GCS): typically, 8.0 g glucose (D-(+)-glucose, 99.5%, Aladdin) was dissolved in 80 mL deionized water. Then, the obtained transparent solution was placed in a sealed 100 mL Teflon-lined stainless-steel autoclave and heated to 160 °C for 8 h. After cooling naturally to room temperature, the precipitate was separated from the supernatant by centrifugation at 10,000 r/min for 10 min. The resulting solid was purified three times with deionized water and, finally, once with absolute ethanol. The obtained product was vacuum dried at 80 °C overnight. Finally, the dry powder was calcined at 200 °C for 12 h in a muffle furnace to obtain the glucose-derived carbon spheres precursor (p-GCS), which was then carbonized at 500 °C in an argon atmosphere for 2 h to obtain glucose-derived carbon spheres (GCS).

Synthesis of p-GCS@H-PAN and H-PAN: generally, 0.1 g of polyacrylonitrile (PAN, M_w_ = 150,000) was dissolved in 3 mL of N, N-dimethyl formamide (DMF), followed by the addition of 0.05 g of p-GCS. The suspension endured ultrasonic treatment for 2 h to fully disperse p-GCS. A total of 3 mL glycerin and 30 mL isopropyl alcohol were stirred at 500 rpm for 1 h in a 50 mL Teflon polytetrafluoroethylene (PTFE) beaker. Next, the speed was increased to 700 rpm and the as-prepared DMF suspension was added dropwise into the mixed alcohol system. After stirring for another 10 min, the reactor was transferred into 180 °C electrical oven for 6 h. The resulting brown precipitate was thoroughly cleaned with deionized water and dried by freeze-drying. The hydrothermally treated PAN coated p-GCS was abbreviated as p-GCS@H-PAN. The H-PAN was prepared by the same procedures as p-GCS@H-PAN but without the addition of p-GCS.

Synthesis of SGCS@NSC, NSC, and SGCS: the above-prepared p-GCS@H-PAN, H-PAN, and p-GCS were mixed with sublimed sulfur in a ratio of 1:5 and heated to 500 °C for 2 h at 3 °C min^−1^ in an argon atmosphere. The obtained black products were labeled as SGCS@NSC, NSC, and SGCS.

Materials Characterization: S4800 Field emission scanning electron microscopy (SEM Hitachi, Tokyo, Japan) and JEM2100 Transmission electron microscope (TEM) tests were used to characterize the microstructure of the samples. The X-ray diffraction (XRD) powder diffraction pattern of the samples was recorded on a D8 ADVANCE DAVINCI X-ray diffractometer (Cu k_α_ radiation, λ = 0.154 nm BRUKER AXS, Bremen, Germany). Axis Ultra DLD X-ray photoelectron spectroscopy (XPS, Kratoms Britain) were used to analyze the chemical composition of the samples. The structure of carbon component of the prepared samples was characterized by Renishaw India Reflex confocal Raman microscopy (Renishaw, Kingswood, UK). ASAP 2020 surface area and a porosity analyzer were used to analyze the pore structure of the sample.

Electrochemical measurements: The synthesized products were mixed with a conductive carbon (Super P) and sodium carboxymethyl cellulose (CMC) binder with a mass ratio of 8:1:1 in deionized water. The electrodes were fabricated by spreading the slurry on Cu foil and then dried at 80 °C in a vacuum overnight. Whatman glass fiber was used as the separator, sodium metal foil as the counter electrode, and 1 M NaPF_6_ dissolved in ethylene (EC) and dimethyl carbonate (DMC) (1:1, vol/vol) as the electrolyte. The Cyclic Voltammetry (CV) curves were obtained by an electrochemical workstation (CHI 660E) at different sweep speeds between 0.01 and 3.0 V (vs. Na^+^/Na). On the LAND instrument, the constant current charge and discharge experiments with different current densities were performed in the same voltage range. Electrochemical Impedance Spectroscopy (EIS) tests were conducted on an electrochemical station (Model 1470E multi-channel electrochemical workstation, Solartron Metrology) in the frequency range of 100 kHz to 10 mHz at room temperature.

## 4. Conclusions

In summary, glucose-derived carbon spheres were encapsulated into a N, S co-doped 3D cross-linked network for the construction of high-performance sulfur-containing anode for sodium-ion batteries. The tight coating relationship of the NSC layer on SGCS acting as a physical barrier and abundance of C-S/C-N bonds serving as chemical adsorption sites together protect SGCS@NSC from the shuttling effect of polysulfides. In addition, the doped sites can provide additional sodium storage sites and improve the sodium diffusion coefficient of the composite to ensure the comprehensive electrochemical performance of SGCS@NSC. Thus SGCS@NSC still delivers 293.7 mAh g^−1^ sodium storage specific capacity after 200 cycles at 0.5 A g^−1^, higher than 107.4 mAh g^−1^ of SGCS, demonstrating improved cycling stability. With the assistance of a 3D high-speed electron and ion transport network of the NSC layer, SGCS@NSC illustrates enhanced pseudocapacitive charge storage and rate performance. SGCS@NSC provides a high specific capacity of 248.9 mAh g^−1^ at 6.4 A g^−1^ and the pseudocapacitive contribution, of which is as high as 90% at 1.0 mV s^−1^. This multifunctional coating greatly improves the electrochemical performance of SGCS and is also applicable to other sulfur-containing electrodes.

## Figures and Tables

**Figure 1 molecules-28-03335-f001:**
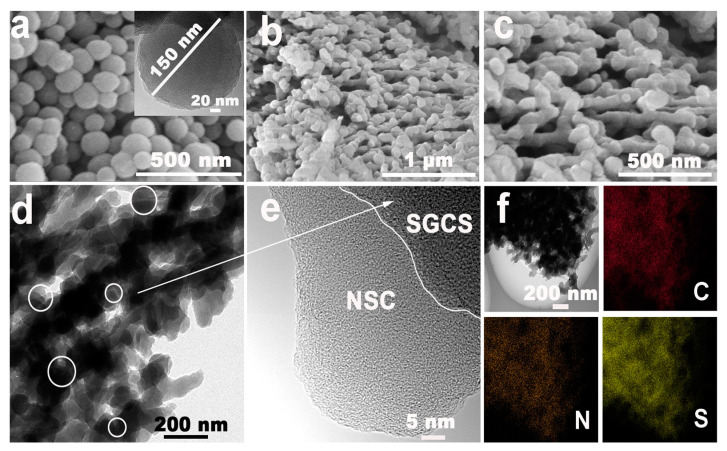
(**a**) The SEM and TEM images of GCS. (**b**,**c**) The SEM images of SGCS@NSC. (**d**,**e**) The TEM images of SGCS@NSC. (**f**) Element mapping images (C, N, and S) of SGCS@NSC.

**Figure 2 molecules-28-03335-f002:**
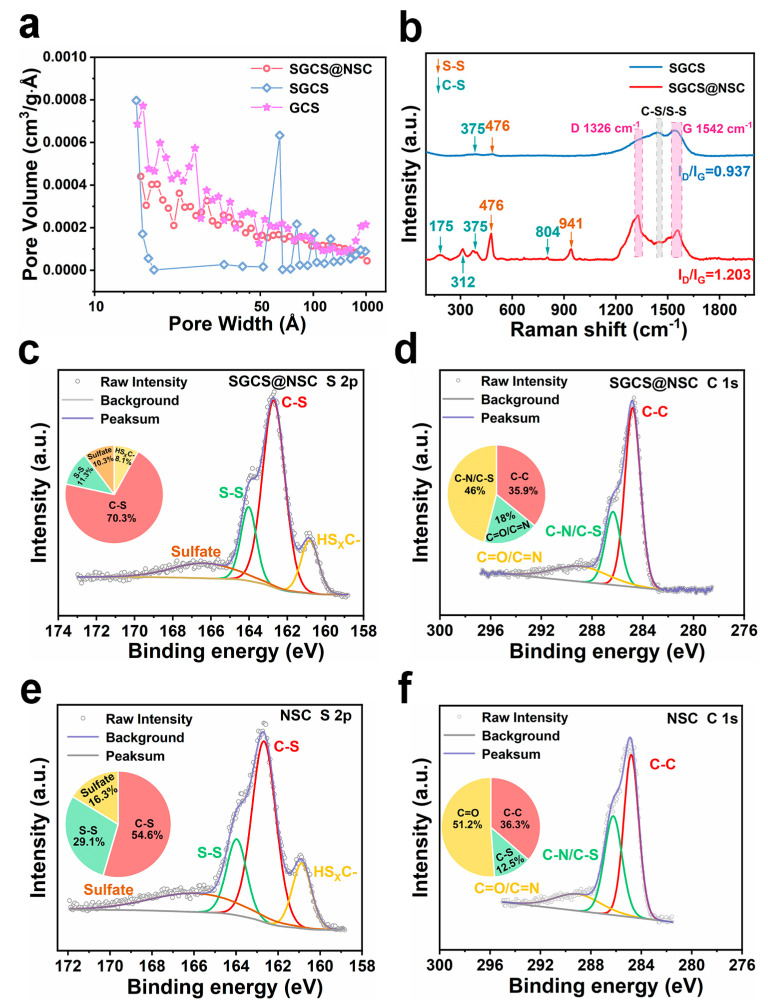
(**a**) Pore distribution curves of SGCS@NSC, SGCS, and GCS. (**b**) Raman scattering spectra of SGCS@NSC and SGCS. (**c**–**f**) The high-resolution XPS of S 2p and C 1s of SGCS@NSC and SGCS.

**Figure 3 molecules-28-03335-f003:**
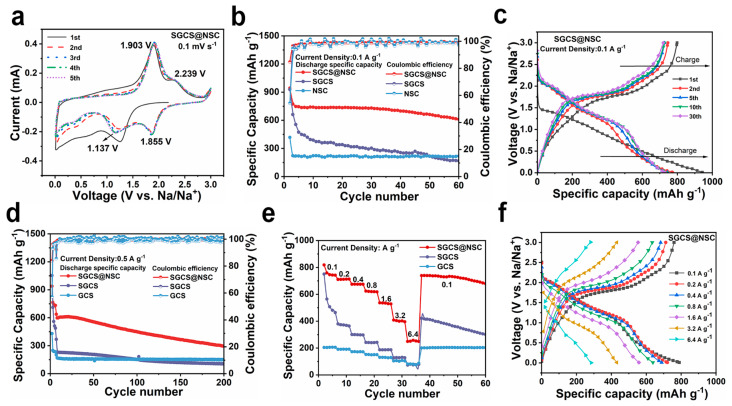
The electrochemical characterization of SGCS, SGCS@NSC, and NSC in the voltage range of 0.01–3.0 V versus Na/Na^+^. (**a**) The first five CV curves of SGCS@NSC at 0.1 mV s^−1^. (**b**) Cycling test at 0.1 A g^−1^. (**c**) The 1st, 2nd, 5th, 10th, and 30th cycles discharge–charge profiles of SGCS@NSC at 100 mA g^−1^. (**d**) Cycling test at 0.5 A g^−1^. (**e**) Rate capability electrodes at increasing current densities from 0.1 to 6.4 A g^−1^. (**f**) Galvanostatic discharge–charge profiles of SGCS@NSC at increasing current densities from 0.1 to 6.4 A g^−1^.

**Figure 4 molecules-28-03335-f004:**
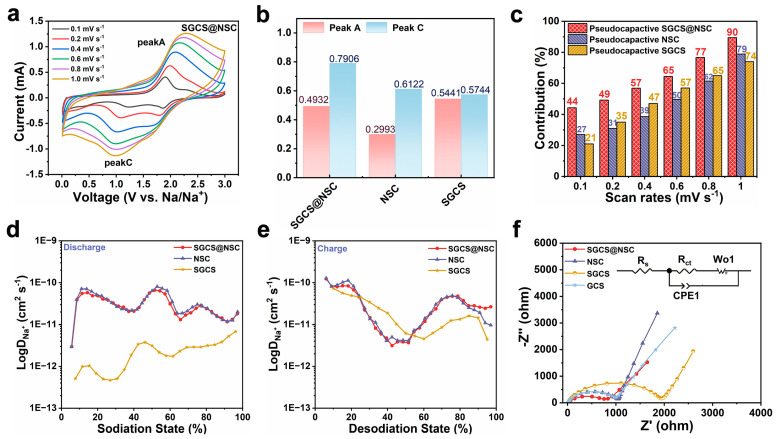
(**a**) CV curves of SGCS@NSC with scan rates from 0.1 to 1 mV s^−1^. (**b**) Log (i) versus log (v) plots of SGCS@NSC, NSC, and SGCS at peak A and C. (**c**) Normalized pseudocapacitive contribution ratios of SGCS@NSC, NSC, and GCS at different scan rates (0.1–1 mV s^−1^). (**d**,**e**) The Na^+^ diffusion coefficient of SGCS@NSC, SGCS, and NSC (discharge and charge) obtained by GITT tests. (**f**) EIS spectra of SGCS@NSC, NSC, SGCS, and GCS electrodes and equivalent circuit.

## Data Availability

The data presented in this study are available in the Appendix A.

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
