# Peer review of "A Multifunctional Coating on Sulfur-Containing Carbon-Based Anode for High-Performance Sodium-Ion Batteries"

_molecules, 2023, doi:10.3390/molecules28083335_

Round 1

Reviewer 1 Report

s

This manuscript reported a SGCS@NSC carbon-based anode material for Na-batteries. The strategy for solving the soluble of  polysulfide intermediates is interesting. There are several points that can potentially help improve the quality of manuscript.

1) How was the value of "17.6%" obtained? It may make no sense by simply use the difference of 46.1% and 28.5%......

2) In the introduction, the author claimed the carbon based materies is most promising for Na-ion storage. However, there is no summary of the current progress of different carbon-based materials and the defination of HC should be given.

3) Figur 2: for calculating pore-distribution, which curve was used, the adsorption or desorption profile?

4) what's the reason of the poor durablity of coated anode )SGCS@NSC) though the initial activity is really high.

5) How about their performance after assembling into battery cell rather than testing in electrochemical cell?

Reviewer 2 Report

The article reports the synthesis and charge storage performance of carbon-based materials for sodium-ion batteries. The best performance was observed for the S, N – co-doped material (SGCS@NSC) with a capacity of 609 mAhg-1 at 0.1 A g-1, and a retention capacity of 46.1% after 200 cycles at 0.5 A g-1. The study supports the current interest in developing materials capable of providing both sodium storage and insertion to improve SIB performances.

It is a well-written article addressing a key topic in the field of energy storage, but there are several issues that the authors need to address before publishing this study.

1. The introduction does not sufficiently emphasize the novelty of the current work. A comparison with other studies on S, N-doped hard carbon materials is strongly recommended. Please see Xu et al, Adv. Energy Mater.2016,6,1501929 “A Hierarchical N/S-Codoped Carbon Anode Fabricated Facilely from Cellulose/Polyaniline Microspheres for High-Performance Sodium-Ion Batteries”; Shao et al, Adv. Energy Sustainability Res. 2022, 3, 2200009.

2. Indicate the interlayer distance and explain the differences, if any, between the synthesized carbonaceous materials. 

3. The exact position of the D and G bands observed in the Raman spectra and the justification for the differences need to be discussed (Fig2b, S2 b,c). Is there any indication related to the contribution of the GCS core and NSC shell to the SGCS@NSC structure/properties?

4. What is the C/N ratio before and after doping with S?

5. The experimental section is not very clear. The method used to purify GCS is missing. Authors are encouraged to rewrite this section to be easily reproduced.

6. The synthesis of the GCS-500 control sample by calcination of the GCS at 500C in the absence of Sulphur is reported but not used in the study. Thus, the authors have to explain the reason behind this experiment.

7. The designation of polyacrylonitrile-coated GCS as GCS@NSC is confusing. Moreover, both polyacrylonitrile and Sulphur-doped polyacrylonitrile are denoted as NSC; see experimental section lines 243,245 234. Choosing different abbreviations for the polyacrylonitrile before and after doping with Sulphur is strongly recommended.

8. The authors should consider changing the font and size of the figures; all figures and insets are too small to read.

Reviewer 3 Report

The article "Multifunctional Coating on Sulfur-containing Carbon-based anode for high-performance sodium-ion batteries." The subject is interesting; however, the manuscript lacks details and clarifications. So I am suggesting the following points to upgrade the manuscript to meet the journal's standards.

1.      The authors present the excellent electrochemical performance of SGCS@NSC compared to the other materials. However, the theoretical explanation is lacking to support their result.

2.      How did the authors control the thickness of the outer carbon layer on the GCS?

3.      The authors should emphasize the role of hetero atoms doped in the carbon.

4.      The EIS  interpretation (Fig. 4 (f)) is very poor in the manuscript. The authors should explain more, including solution and charge transfer resistance.

Round 2

Reviewer 2 Report

The article can be published in its current form, although some issues remain unaddressed. Relevant references have been added to the introduction, but a direct comparison has not yet been completed. The TEM image shows the presence of the amorphous NSC shell around a GCS.